# DeepStableYolo: DeepSeek-Driven Prompt Engineering and Search-based Optimization for AI Image Generation

**Héctor D. Menéndez**
Department of Informatics
King's College London
London, United Kingdom
hector.menendez@kcl.ac.uk

**Gema Bello-Orgaz, Cristian Ramirez-Atencia**
ETSI de Sistemas Informáticos
Universidad Politécnica de Madrid
Departamento de Sistemas Informáticos, Madrid, Spain
{gema.borgaz,cristian.ramirez}@upm.es

## Abstract

AI-driven image generation heavily relies on effective prompt engineering and precise tuning of model parameters. The StableYolo framework addressed these challenges by integrating evolutionary computation with Stable Diffusion, enabling simultaneous optimization of both prompts and model parameters while using YOLO as a guiding metric to enhance image quality. In this work, we extend the capabilities of StableYolo by introducing mechanisms for prompt improvement through large language models (LLMs), aiming to maximize image generation quality. We incorporate DeepSeek to enhance prompt engineering, ensuring more effective and context-aware prompt generation. However, our refined approach demonstrates that enhancing prompts does not yield significant improvements in either the efficiency or quality of AI-generated images, suggesting that clear and concise prompts are equally effective in the process.

## 1 Introduction

AI-based image generation has achieved remarkable progress, with models like Stable Diffusion at the forefront [12]. However, generating high-quality images still depends heavily on two critical aspects: effective prompt engineering and precise tuning of model parameters. Both tasks require significant manual effort and expertise, presenting ongoing challenges for researchers and users alike [14]. The StableYolo framework [1] addressed these challenges by integrating evolutionary computation with Stable Diffusion, simultaneously optimizing prompts and model parameters while using YOLO as a guiding metric to enhance image quality [11].

Building on this foundation, we propose an extension to StableYolo that enhances the optimization process by integrating DeepSeek [7], a state-of-the-art large language model (LLM), for the generation and refinement of both the prompt and the negative prompts. DeepSeek's advanced natural language reflective capabilities enable the automatic generation of context-aware and semantically rich prompts, significantly reducing the need for manual intervention [3]. Meanwhile, StableYolo's search process focuses on optimizing the remaining parameters of Stable Diffusion, such as guidance scale, inference steps, and other hyperparameters, ensuring a comprehensive and automated tuning process [12].

This refined approach aims to enhance the efficiency and quality of AI-generated images by improving the exploration of the search space through optimized prompt descriptions. By combining DeepSeek's prompt-enhancing capabilities with the optimization power of search algorithms, we aim to evaluate whether original prompts can be improved and how this impacts the final quality of the generated images. However, during our experiments, our results showed that enhancing prompts with DeepSeek

XVI XVI Congreso Español de Metaheurísticas, Algoritmos Evolutivos y Bioinspirados (maeb 2025).

does not yield any noticeable improvements in StableYolo's search process. The main contributions of this extended work are:

1. DeepStableYolo, which integrates DeepSeek into the prompt engineering process of StableYolo for enhanced generation of positive and negative prompts.[1]

2. A comprehensive analysis to evaluate whether enriching prompts with LLMs can expand the search space and reveal unexplored areas in the image generation process, while also addressing the associated limitations and trade-offs.

3. The identification of the most frequently used words in prompts that enhance image quality, offering insights into how prompt structure influences model performance.

## 2  Background

Recent advancements in artificial intelligence have led to the development of powerful image generation models capable of creating high-quality images from textual descriptions, revolutionizing fields such as graphic design, advertising, and entertainment [10, 12]. However, achieving optimal results still depends on two critical factors: effective prompt engineering and precise parameter tuning. Both require significant expertise and manual effort [2].

To enhance text-based image generation, various prompt engineering techniques have been explored. Large language models (LLMs) like GPT-3 [3] and DeepSeek [8] have been utilized to generate more coherent and contextually relevant prompts, thereby improving image generation outcomes. While these advancements are promising, integrating LLMs into image generation workflows presents challenges related to scalability and computational efficiency [6].

A promising approach to improving AI-driven image generation involves combining genetic algorithms with LLMs to optimize both prompts and model parameters. For example, the Multimodal LLM Adapter (MoMA) framework [13] introduces a multimodal LLM adapter for personalized image generation, synergizing reference images and text prompts to produce high-quality images. Additionally, iterative prompting techniques using multimodal LLMs have been employed to reproduce both natural and AI-generated images, demonstrating the potential of integrating evolutionary algorithms with LLMs for prompt engineering [9]. These approaches aim to automate and enhance the image generation process while balancing factors such as image quality and computational efficiency.

Models like DeepSeek [7] have demonstrated advanced natural language capabilities, making them well-suited for automating and improving the prompt engineering process. Their integration into image generation workflows not only reduces manual intervention but also enables the creation of semantically richer and contextually relevant prompts. Furthermore, combining StableYolo [1] with these advanced LLMs offers a significant advantage by optimizing multiple aspects of the image generation process simultaneously. By balancing objectives such as image quality, computational efficiency, and semantic relevance, this combination should, in principle, provide a more refined and automated workflow. However, this is not always the case, as optimization does not necessarily require additional boosting to create suitable prompts for generating high-quality images, as we will see in Section 5.

## 3  Methodology

With a user-provided prompt for generating photo-realistic images, this work aims to enhance image quality through evolutionary search. Each individual in the population is represented as a dictionary object that incorporates parameters based on StableDiffusion's documentation [5]. The structure of each individual includes the following key parameters:

- **Number of Iterations**: The number of diffusion steps required for the AI to process the image (range: [1, 100]).
- **Classifier-Free Guidance Scale (CFG)**: A parameter that controls the influence of the prompt on the generated image. Higher values increase the prompt's influence but may also reduce image quality if set too high (range: [1, 20]).

---

[1]The code (https://github.com/hdg7/stableyolo) and data (https://zenodo.org/records/14933760) are openly available.

- **Seed**: The generation seed used for randomization. It ensures consistent image generation when using the same seed or variation with different seeds. It is included in the search to guarantee consistency between genotypes and fenotypes.
- **Guidance Rescale**: A parameter that prevents overexposure by rescaling the guidance factor (range: [0, 1]).
- **Positive Prompt**: The text or set of words describing the desired image and its details, aimed at enhancing the realism of the generated image.
- **Negative Prompt**: A sequence of keywords to be excluded during image generation to reduce non-realistic components in the final output.

DeepSeek is employed to refine both the positive and negative prompts, producing more contextually relevant and semantically enriched text that enhances the realism of the generated images. This approach extends the original StableYolo engine, which solely employs a search process to identify suitable prompts [1]. The prompt used to enhance the positive and negative prompts for StableDiffusion is as follows:

```
" The following prompt aims to generate a good quality photograph from
    StableDiffussion. It is a < positive | negative > prompt. Please
    provide an enhanced version as a string starting with ```txt and
    ending with ```. The text is: "
```
Listing 1: Enhanced Prompt

The rest of the prompt consists of a list created by StableYolo after selecting the prompt keywords. An example of an individual in the population could be represented as:

```
{
    'iterations': 50,
    'cfg_scale': 15,
    'seed': 12345,
    'guidance_rescale': 0.5,
    'positive_prompt': 'a person, photograph, digital, color, blended visuals',
    'negative_prompt': 'illustration, painting, drawing, art'
}
```

As presented in Algorithm 1, the image generation process follows an evolutionary framework, where the population aims to optimize image quality using YOLO as the evaluation metric. Initially, the algorithm creates the population by setting parameters at random, defining a user-specified prompt goal (e.g., "a person"), and selecting prompt variations from a fixed list of topics. Once these variations are chosen, DeepSeek is employed for prompt enrichment each time a new individual is generated. The population then evolves through a genetic algorithm that applies extended crossover and mutation operators.

The crossover operator facilitates the exchange of values between two individuals by selecting which values to swap uniformly at random. Mutation modifies these values within their allowable ranges. Specifically, for the prompts, mutation selects alternate word sets from the predefined vocabulary available for both positive and negative prompts.

The YOLO_score, utilized as part of the fitness evaluation process, is calculated through four distinct steps:

1. **Prompt Generation and Configuration**: StableYolo generates both positive and negative prompts. These prompts are then used to configure the parameters of Stable Diffusion, resulting in the production of four images per prompt.
2. **Prompt Enhancement for Photorealism**: DeepSeek enhances these generated prompts with the objective of producing photorealistic images.
3. **Object Detection and Confidence Scoring**: YOLO processes each image, detecting objects within them. For each detection, a confidence score is assigned, which serves as an indicator of the image's quality.
4. **Final Score Calculation**: The final value is determined by averaging these confidence scores across all detected objects and generated images.

**Algorithm 1** DeepStableYolo algorithm

---

**Require:** Population size $N$, number of generations $G$, crossover rate $p_c$, mutation rate $p_m$, the original user prompt $prompt$
**Ensure:** Optimised model parameters and prompts variations
1:   $P \leftarrow$ CreateRandomPopulation$(N)$          ▷ Initialize population
2:   $Fitness \leftarrow$ YOLO_score$(P)$          ▷ Evaluate YOLO score
3:   **for** $t = 1$ to $G$ **do**          ▷ Main loop
4:      $parents \leftarrow$ SelectionTournament$(N, P)$          ▷ Select parents
5:      $offspring \leftarrow \emptyset$          ▷ Initialize offspring set
6:      **for all** $p_1, p_2 \in parents$ **do**          ▷ Generate offspring
7:         $child \leftarrow$ CrossoverAndMutation$(p_1, p_2, p_c, p_m)$
8:         $child.prompts \leftarrow$ DeepSeekEnhace$(child.prompts)$          ▷ LLM-based prompt improvement
9:         $offspring \leftarrow offspring \cup \{child\}$
10:      **end for**
11:     $Fitness \leftarrow$ YOLO_score$(offspring)$          ▷ Evaluate YOLO score for offspring
12:     $P \leftarrow$ Replacement$(P, offspring)$          ▷ Replace population using $\mu + \lambda$
13: **end for**
14: **return** Population $P$          ▷ Return final solutions

---

Table 1: Genetic parameters of DeepStableYolo algorithm, following the original parameters of StableYolo [1]

| Name | Description | Value ($\mu$m) |
|------|-------------|----------------|
| N | Population size | 25 |
| G | Maximum number of generations | 50 |
| $\mu$ | Number of individuals to select for the next generation | 5 |
| $p_c$ | Crossover probability | 0.2 |
| $p_m$ | Mutation probability | 0.2 |

## 4   Experimental Setup

To evaluate the new approach for the StableYolo framework thoroughly, we utilized 10 distinct categories of objects, animals, and people recognized by YOLO (concretely: banana, bear, bird, cat, dog, elephant, giraffe, train, person, and zebra). These categories were selected to investigate how prompts influence achieving optimal image quality. To gauge improvements, we compared the baseline prompt with enhanced (DeepStableYolo) and StableYolo's prompts. Our experiments aimed to address the following research questions:

**RQ1**: Do StableYolo and DeepStableYolo enhance baseline results?

**RQ2**: What are the most common words in the prompts, and what improvements do they bring compared to DeepStableYolo?

**RQ3**: How do images differ between StableYolo and DeepStableYolo?

**RQ4**: Do prompts reduce the need for other parameters, such as inference steps?

The genetic algorithm (GA) settings are outlined in Table 1. Each experiment was repeated four times to ensure reliability. All experiments were conducted on a workstation with Ubuntu 20.04 LTS, equipped with 40 CPU cores, 128 GB RAM, and an NVIDIA A30 GPU featuring 24 GB memory.

## 5   Results

Our aim is to investigate whether enhanced prompts can improve image generation outcomes using StableYolo. To accomplish this, we integrated DeepSeek into our prompt enhancement process. DeepSeek enriches the text of the prompt by extending its length and encouraging broader exploration, enabling us to uncover areas of the search space that were previously unexplored with StableYolo's

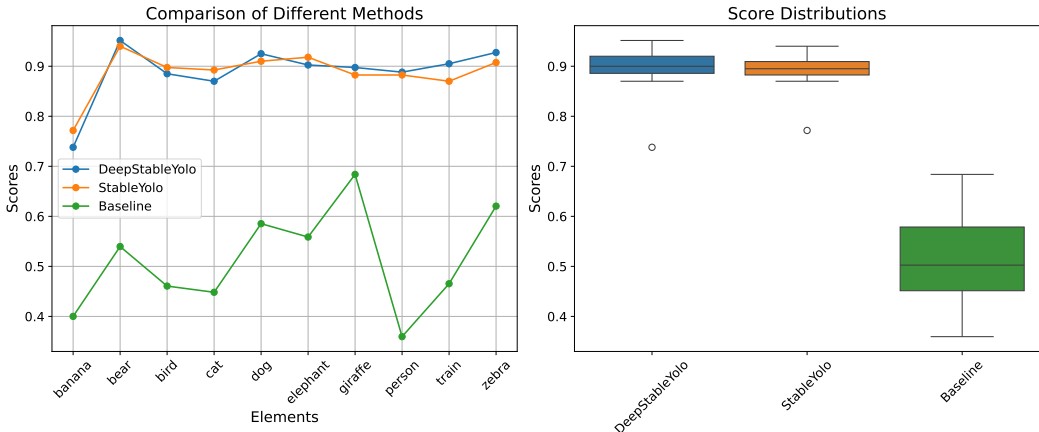

Figure 1: Comparitive Results among StableYolo, DeepStableYolo and the baseline values from random parametrizations.

limited prompts. However, this exploration expands the search space in unforeseen ways, providing minimal control over the prompt beyond a baseline level. With this objective in mind, we present the experimental results conducted to address each research question outlined in this study.

## 5.1 RQ1: Comparative Results

To address our first research question, we compare StableYolo and DeepStableYolo. For a fair comparison, both techniques utilize identical parameters as those used in the original evaluation of StableYolo [1]. Additionally, we establish a baseline by presenting results from random parameterizations adhering to the same guidelines.

Figure 1 demonstrates that both techniques enhance the baseline performance. DeepStableYolo achieves an average fitness population of $0.889 \pm 0.055$ across the 10 different cases, while StableYolo attains an average fitness population of $0.887 \pm 0.043$. The baseline, however, yields a significantly lower average fitness of $0.512 \pm 0.097$.

The results vary depending on the image category. For instance, the 'banana' category produces the poorest outcomes (approximately 0.72 for StableYolo, 0.77 for DeepStableYolo, and 0.4 for the baseline), whereas the 'bear' category shows the best performance (0.93 for DeepStableYolo and 0.95 for StableYolo). The remaining categories exhibit stable results around 0.9 for both techniques. In contrast, the baseline results are more variable, with image quality scores ranging from 0.7 to 0.35.

Interestingly, no significant improvements in image generation were observed between StableYolo and DeepStableYolo. Applying the Wilcoxon test [4] to their image quality scores resulted in a p-value of 0.90, which is considerably higher than the 0.05 threshold required to reject the null hypothesis. Thus, we cannot consider these results significantly different. Regarding convergence points, StableYolo converges on average at generation $26.3 \pm 15.09$, while DeepStableYolo converges at $30.1 \pm 12.26$. The Wilcoxon test revealed no statistically significant difference between their convergence points (p-value = 0.65). These findings suggest that enhancing prompts does not significantly influence either the search quality or the convergence performance of the approaches. Furthermore, despite increasing the search space by enhancing prompts, the existing search space for image generation is already sufficient to yield highly competitive results.

## 5.2 RQ2: Prompt Comparison

DeepSeek uses the prompt created by StableYolo to develop an enhanced prompt. This enhancement may introduce specific terms that can improve image quality, which were not considered in StableYolo's design. Additionally, it improves prompt text quality by ensuring the LLM generates coherent prompts and applies enhancements based on its knowledge –similarly to Yao et al.'s work [15].

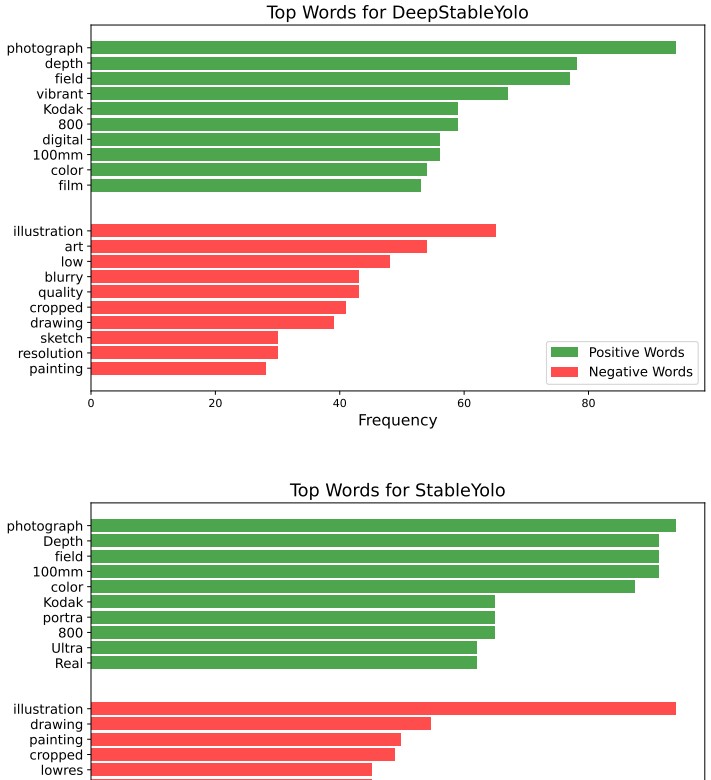

Figure 2: Most frequent words for DeepStableYolo (top) and StableYolo (bottom) according to the top 10 solutions for each category search. The green bars represent the positive prompt while the red ones represent the negative prompts.

The average positive prompt length is $45.0 \pm 21.7$ for DeepStableYolo and $14.86 \pm 2.45$ for StableYolo. For negative prompts, the averages are $21.6 \pm 15.5$ for DeepStableYolo and $4.4 \pm 1.1$ for StableYolo. The prompt lengths differ significantly between both approaches (positive and negative prompts) with a p-value of $2.30 \times 10^{-15}$, which is less than 0.05.

To analyze the most frequent words in each prompt, we examined the top ten terms for every approach and category. Figure 2 illustrates the frequencies of these terms in their positive and negative prompts. The results show some similarities between the two approaches.

The most frequent word in the positive prompts is 'photograph' for both DeepStableYolo and StableYolo. In StableYolo's case, other common terms include 'depth', 'field', '100mm', and 'color'. For DeepStableYolo, while 'depth' and 'field' are also common, the term 'vibrant' emerges as frequent and is not present in StableYolo's prompts.

For negative prompts, both approaches commonly use the term 'illustration'. However, StableYolo frequently includes it in nearly every case. DeepStableYolo's related terms include 'art', 'low', and 'blurry', whereas StableYolo uses terms like 'drawing', 'painting', and 'cropped,' which are not closely tied to 'illustration.'

## 5.3  RQ3: Image Comparison

The final part of the evaluation involves a comparison of the top images across each category. From a quality perspective, there is no clear distinction in image quality between the two techniques. However, some interesting aspects emerge when examining DeepStableYolo's approach. For instance, as shown in Figure 3, the 'banana' generated by DeepStableYolo appears mature, and the 'elephant' image is rendered in grayscale. Additionally, the 'bird' image captures its reflection in the water.

When comparing the 'bear' and 'zebra' images, both renderings are equally plausible for their respective prompts and would fit well in a photorealistic context. The 'train' depictions also show similarities: DeepStableYolo presents a modern version, while StableYolo opts for a classical approach. Notably, both images remain quite similar overall. The last comparison involves the image of a 'person'. DeepStableYolo generates an older individual with distinct facial expressions, whereas StableYolo's output features a younger person depicted from a distance. DeepStableYolo's rendition is significantly more detailed in this case.

Examining the prompts used by DeepStableYolo for these images (see Listing 2, Listing 3, Listing 4, and Listing 5), we observe variations from the original prompts created by StableYolo. Although the language model enhances narrative depth, this does not significantly impact the overall quality of the images. The most notable difference is in the prompt for the person's image, where DeepStableYolo specifies "detailed facial expressions," directing the generation process to focus on the individual's face.

```
Positive: a vibrant banana laid on a rustic wooden table under soft
    golden light, with the warm glow casting long shadows across the
    surface, creating a lively and inviting scene
Negative: a highly stylized detailed illustration that is distorted,
    cartoonish, overly exaggerated, poorly drawn, pixelated, low-
    quality, blurry, with a lot of artifacts or errors in the image
    layout, cropped in error mode, and lacks sharpness.
```
Listing 2: DeepStableYolo prompt for the best image of 'banana'.

```
Positive: bird, photograph, color, Kodak portra 800, Depth of field
    100mm, overlapping compositions, blended visuals
Negative: high quality; illustration; drawing; art; vector art; lowres
    ; error; cropped; sharp focus; detailed; clean; clear; high
    contrast; well-composed; artistic style; minimal distractions;
    precise details; natural lighting; modern aesthetic; elegant;
    professional artwork; fine details; intricate design; refined;
    polished
```
Listing 3: DeepStableYolo prompt for the best image of 'bird'.

```
Positive: a vibrant color palette with warm tones, person, photograph,
     Kodak portra 800 film, depth of field 100mm, soft lighting,
    detailed facial expressions, cinematic blur, overlapping
    compositions with smooth transitions between layers, ethereal
    quality, and a sense of depth and detail in the shadows
Negative: txt:illustration,drawing,art,fake,synthetic,blurry,cropped,
    error,negative image,lowres,highly unrealistic,bad quality
```
Listing 4: DeepStableYolo prompt for the best image of 'person'.

```
Positive: zebra, photograph, Ultra Real, Depth of field 100mm,
    trending on artstation, award winning
Negative: High-quality photograph illustration, painting style with
    sketches, realistic artistic vision
```
Listing 5: DeepStableYolo prompt for the best image of 'zebra'.

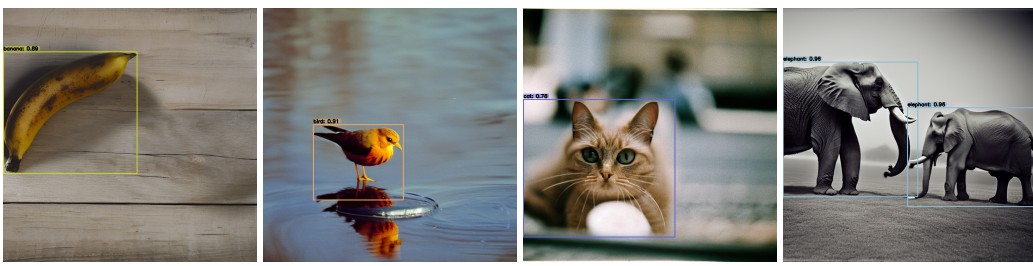

(a) Prompts Enhanced with DeepSeek.

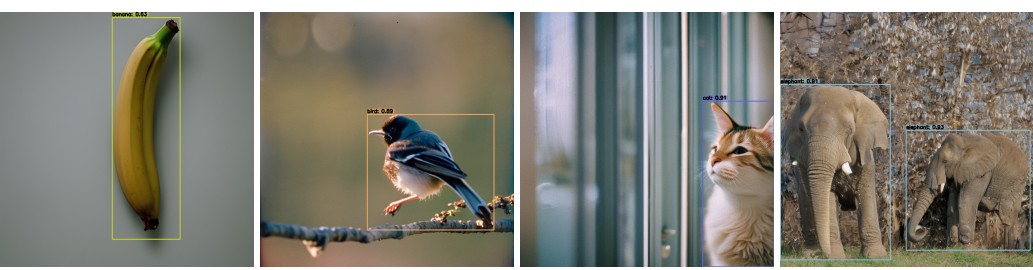

(b) Original StableYolo Prompts.

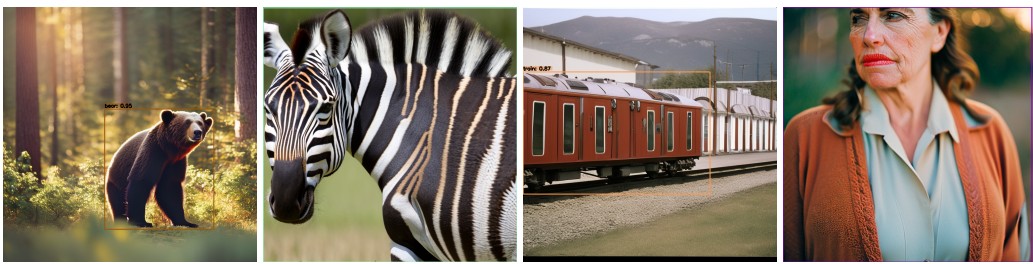

(c) Prompts Enhanced with DeepSeek.

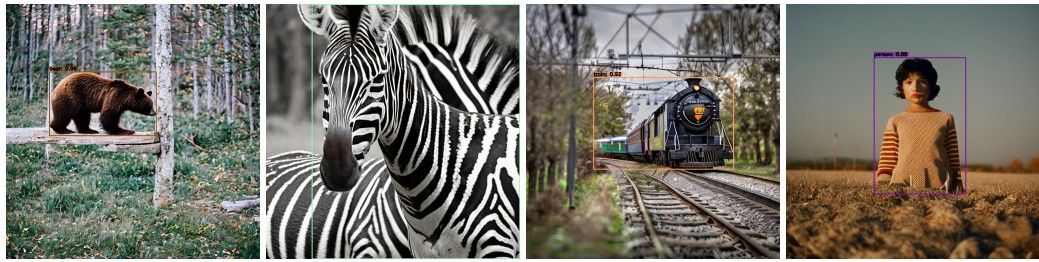

(d) Original StableYolo Prompts.

Figure 3: Comparison of image differences between StableYolo and DeepStableYolo.

### 5.4 RQ4: Parameters Comparison

The remaining parameters generated during the search processes of DeepStableYolo and StableYolo also exhibited no significant differences.

The first parameter is 'Inference Steps'. For DeepStableYolo, the mean value was $50.5 \pm 22.95$, while for StableYolo, it was $47.2 \pm 25.54$. After applying the Wilcoxon test to both distributions, the p-value was 0.92, showing no statistical difference between them.

For the 'Guidance Scale', the values were also similar. For DeepStableYolo, the mean value was $10.7 \pm 2.47$, and for StableYolo, it was $9.5 \pm 2.18$. The p-value for the Wilcoxon test was 0.32, indicating no statistically significant difference between the two distributions.

For the 'Guidance Rescale', the average values were $0.43 \pm 0.21$ for DeepStableYolo and $0.68 \pm 0.17$ for StableYolo. After applying the Wilcoxon test, the p-value was 0.04, indicating a statistically significant difference between the two groups. However, this parameter does not affect generation performance in practice, similar to how 'Inference Steps' or the 'Guidance Scale' influence the process.

## 6 Discussion and Conclusions

In general terms, the study demonstrates that while LLM-enhanced prompts can enrich the narrative and expand the search space for image generation, they do not significantly improve image quality compared to methods like StableYolo where promps are chosen manually.

However, the findings of this study have significant implications for the field of prompt engineering and image generation. While models such as DeepSeek can enhance prompts and expand the search space, this does not necessarily result in improved image quality, suggesting that the effectiveness of prompt enhancement is context-dependent and requires further investigation. Moreover, the similarity in word frequency between StableYolo and DeepStableYolo indicates that certain core elements, such as "photograph" and "depth," are essential for high-quality image generation, highlighting the potential for their optimization in future research. Additionally, although the creative modifications introduced by DeepStableYolo enhance visual appeal, they do not lead to measurable improvements in image quality, raising important questions about the trade-offs between creativity and technical performance in generative models.

Based on the findings of this study, several avenues for future research can be identified. Further exploration of alternative prompt enhancement techniques could be valuable, including domain-specific fine-tuning of LLMs or integrating user feedback into the prompt generation process. Additionally, while DeepStableYolo produces longer prompts, the relationship between prompt length and image quality remains uncertain, warranting further investigation into whether an optimal length exists for different image generation tasks. Moreover, as this study focused on a specific set of image categories, future research could extend the evaluation to a wider range of domains to assess the broader applicability of these findings.

## Acknowledgments

The support of the UKRI Trustworthy Autonomous Systems Hub (reference EP/V00784X/1), Trustworthy Autonomous Systems Node in Verifiability (reference EP/V026801/2) and the Comunidad Autónoma de Madrid under ALENTAR-J-CM project (reference TEC-2024/COM-224) is gratefully acknowledged.

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
