# OpenReview forum: "DeepStableYolo: DeepSeek-Driven Prompt Engineering and Search-based Optimization for AI Image Generation"
_MAEB/2025/Congreso — MAEB 2025_

### Official Review · Reviewer_f661 · 2025-03-06
**Good paper with "disappointing" results**

**Rating:** 5
**Confidence:** 4

**Review:**

This paper presents a extension of StableYolo by expanding the prompts with DeepSeek. StableYolo improves the quality/realism of generated images (as measured by YOLO) automatically with prompt engineering via metaheuristics. This work extends SableYolo by enriching the image generation prompts with the LLM DeepSeek. The results suggest that enriching the prompts with SatableYolo brings no measurable improvement in performance.

I think this is well conducted work. The community could benefit from knowing that there might not be further improvement in this direction.
1) One doubt that I have is why do we need to use a LLM like DeepSeek? Would it not be better/simpler to use word2vec to get similar /close by words and enrich it like that? For example, a "mutation" could be changing one word into another close by word.

2) I disagree with the future directions:

"""
Based on the findings of this study, several avenues for future research can be identified. Further exploration of alternative prompt enhancement techniques could be valuable, including domain-specific fine-tuning of LLMs or integrating user feedback into the prompt generation process. Additionally, while DeepStableYolo produces longer prompts, the relationship between prompt length and image quality remains uncertain, warranting further investigation into whether an optimal length exists for different image generation tasks. Moreover, as this study focused on a specific set of image categories, future research could extend the evaluation to a wider range of domains to assess the broader applicability of these findings.
"""
I respectfully disagree. I think this work shows that perhaps there is not much improvement to do in this direction.


Overall, I think this is a well conducted research, even if the results are disappointing.

---

### Official Review · Reviewer_7wW1 · 2025-03-08
**Paper is interesting but shows tension between what the authors expected and what they found**

**Rating:** 4
**Confidence:** 4

**Review:**

The paper is straightforward. The authors are evaluating the integration of Deep Seek into an image generation model called StableYolo, that combines StableDiffusion as the core generative AI, a genetic algorithm to tune hyperparameters and search for an optimal prompt, and YOLO for fitness computation. DeepSeekl is used to tune the prompt, with the hypothesis that the LLM can improve the quality of the generated images by improving the prompt. Experiments show that while StableYolo, and the DeepSeek version both outperform the baseline, they are basically equivalent based on the evaluation criterion, and qualitative comparisons. The work is interesting and results are relevant, the paper should be accepted.

However, the manuscript should be improved on several points. While most issues are minor or specific, there is one larger issue that is palpable in all of the manuscript. There is a tension between what the authors expected to find, and what they did find in their results. In the paper they describe the DeepSeek LLM as being able to "enhance the efficiency and quality of AI-generated images by improving the exploration of the search space through optimized prompt descriptions", this by introducing "creative modifications" but "results proved that enhancing prompts with DeepSeek does not yield any noticeable improvements". The authors claim that DeepSeek "enriches prompts", and that "LLM-enhanced prompts can enrich the narrative and expand the search space for image generation, " but that "they do not significantly improve image quality". In what way then, did DeepSeek "enrich prompts"? It seems that the authors are confusing their own human interpretation of the prompts with what the prompts actually contain and how they are processed by StableDiffussion as inputs. My suggestion is that the authors should take a more objective approach to how they describe their work, hypothesis, experimental methodology, and especially their assumptions and claims concerning their work.

Other issues:

- Not all acronyms are defined, some are redefined several times, and some are not used consistently
- both "Prompt and PromptNeg (negative prompts)" why capitalize the words, and PromptNeg is only used here,
- While LLMs like DeepSeek can produce impressive results, stating that they possess "understand" language is an over statement to put it mildly
- "our results proved that .." did you really prove something?
- "GPT-3 [3] and DeepSeek [8] have been utilized to generate more coherent and contextually relevant prompts," ... so previous works show that in fact they might be able to produce the results aimed for in this work, but your results suggest the contrary. This is interesting, I think a deeper discussion of what previous works have found, and how your results contrasts with them, is sorely missing in the manuscript
- "more refined ... workflow." very subjective
- "With a user-provided prompt for generating photo-realistic images, this work aims to enhance image quality through evolutionary search." but this is not what is shown in THIS work, that was in previous works
- (Range: [1, 20]). ) ... why use this format? why is range in capitalized
- Why is the random seed optimized? this seems odd, I think you cannot justify this choice
- The caption of most figures and listings are missing a closing period
- Numerical results presented in Section 5 should be put in a table (see for instance Section 5.4, but others as well, it is very difficult to read)
- IN figure 4, also include the baseline results
- Figure 2 and 3 should be combined, at least as side by side subfigures, but overlayed would be better, overlapping on the words used in both
- Seeing Figure 4, I see no improvement from DeepSeek in general,

---

### Official Review · Reviewer_RVFC · 2025-03-19
**Interesting contribution where an LLM (DeepSeek) is incorporated to an evolutionary framework to design realistic images using Stable Diffusion and Yolo**

**Rating:** 5
**Confidence:** 5

**Review:**

This is an interesting contribution extending the StableYolo existing proposal in [1]. StableYolo uses a genetic algorithm to both optimize prompts and Stable Diffusion parameters to automatically generate realistic images, while Yolo is considered in the fitness function to measure the image quality by detecting the objects in the generated image. The current proposal incorporates an LLM stage to the StableYolo cycle based on DeepSeek to optimize the generated prompts. The proposal is well designed, the text is carefully written, the state of the art is perfectly reviewed and the experimental setup is sound. Even if the results obtained are not the expected since the DeepStableYoloproposal does not improve the performance of StableYolo, the manuscript constitutes a nice research exercise and deserves to be accepted for publication in the conference.

Below are some minor issues:

1) I wonder why the seed for Stable Diffussion is included in the genetic algorithm's coding scheme. Although this decision comes from the original StableYolo proposal, I think it is not a good design choice as the exploration space of seeds is huge and it would be difficult to find a good value. My view is that it would be better to fix the seed. The authors can discuss this issue in the text of the revised version.

2) The description of the genetic algorithm design is generally good but the replacement mechanism (step 12 in Algorithm 1) is neglected. Which mechanism is considered? This is even more important as you include a mu parameter in Table 1 that seems to be related to this mechanism and the readers are not able to understand. Please, include a description of the selection mechanism in the text.

3) Why is the value of the crossover probabilty so low (0.2)? This is not a usual choice for a genetic algorithm. Please, justify this choice.

4) Reference [6] has been already published and it is missing some data: Dec. 2024, pp. 5873-5893, vol. 5

---

### Decision · Program_Chairs · 2025-03-20

Accept